# Resistance to Diet Induced Visceral Fat Accumulation in C57BL/6NTac Mice Is Associated with an Enriched Lactococcus in the Gut Microbiota and the Phenotype of Immune B Cells in Intestine and Adipose Tissue

**DOI:** 10.3390/microorganisms11092153

**Published:** 2023-08-25

**Authors:** Samnhita Raychaudhuri, Md Shahinozzaman, Si Fan, Opeyemi Ogedengbe, Ujjwol Subedi, Diana N. Obanda

**Affiliations:** Department of Nutrition and Food Sciences, University of Maryland, College Park, MD 20742, USA; samnhita@umd.edu (S.R.); mshahin@umd.edu (M.S.); sfan9@umd.edu (S.F.); ogeds20@umd.edu (O.O.); usubedi@umd.edu (U.S.)

**Keywords:** obese prone, obese resistant, gut bacteria, Lactococcus, Lactobacilli, B1b IgM^+^ cells, IgA^+^ cells

## Abstract

**Highlights:**

**What are the main findings?**
An enriched population of Immune B cells subtypes particularly IgA^+^ cells in the lamina propria and B1B IgM^+^ in adipose tissue may contribute to resistance to diet induced obesity and insulin resistance.Lactobacillus-enriched individuals had a higher probability of diet-induced obesity and insulin resistance while Lactoccocus -enriched individuals had a higher resistance to diet-induced obesity and insulin resistance.

**What is the implication of the main finding?**
Interventions based on diet or gut microbiota to promote the proliferation of these B cell populations in Intestine and in adipose tissue are a potential target for protecting against diet induced obesity.

**Abstract:**

Humans and rodents exhibit a divergent obesity phenotype where not all individuals exposed to a high calorie diet become obese. We hypothesized that in C57BL/6NTac mice, despite a shared genetic background and diet, variations in individual gut microbiota function, immune cell phenotype in the intestine and adipose determine predisposition to obesity. From a larger colony fed a high-fat (HF) diet (60% fat), we obtained twenty-four 18–22-week-old C57BL/6NTac mice. Twelve had responded to the diet, had higher body weight and were termed obese prone (OP). The other 12 had retained a lean frame and were termed obese resistant (OR). We singly housed them for three weeks, monitored food intake and determined insulin resistance, fat accumulation, and small intestinal and fecal gut microbial community membership and structure. From the lamina propria and adipose tissue, we determined the population of total and specific subsets of T and B cells. The OP mice with higher fat accumulation and insulin resistance harbored microbial communities with enhanced capacity for processing dietary sugars, lower alpha diversity, greater abundance of Lactobacilli and low abundance of Clostridia and Desulfobacterota. The OR with less fat accumulation retained insulin sensitivity and harbored microbial communities with enhanced capacity for processing and synthesizing amino acids and higher diversity and greater abundance of Lactococcus, Desulfobacterota and class Clostridia. The B cell phenotype in the lamina propria and mesenteric adipose tissue of OR mice was characterized by a higher population of IgA^+^ cells and B1b IgM^+^ cells, respectively, compared to the OP. We conclude that variable responses to the HF diet are associated with the function of individuals’ gut microbiota and immune responses in the lamina propria and adipose tissue.

## 1. Introduction

While diet-induced obesity primarily results from imbalances in energy intake versus energy expenditure, individuals respond differently to excess nutrients or calories due to the polygenic nature of the obese phenotype. Humans and rodents such as the outbred Sprague-Dawley CD rats and C57BL6 mice have all been shown to exhibit this polygenic nature, where not all individuals exposed to a high-calorie diet become obese [1,2]. Differences in the host obesity phenotype and tissue physiology are partly attributed to the composition and function of the microbiota composition [1,2,3]. The gut microbiota may influence host obesity phenotype through direct contact with intestinal cells or indirectly through bacterial metabolites that are absorbed into the epithelial cells and are delivered to other tissues and cells to impact obesity mechanisms which may include host energy harvest, expenditure and storage [3,4].

In two previous studies, we have shown that the abundance of bacteria from class Clostridia (phylum Firmicutes) and specifically genus *Clostridium* is associated with a lean phenotype. Outbred Sprague-Dawley CD rats that have higher abundance of Clostridia remain lean while those with a relatively less abundance of Clostridia will rapidly gain weight when placed on a high-fat (HF) diet [2,5]. An analysis of recent literature comparing the gut microbiota of people in western industrialized countries where obesity is rampant and hunter-gatherer groups in which there are no obesity issues has shown that the abundance of certain members of phylum Firmicutes in non-western populations may play a role in obesity prevalence [6]. A similar observation has been shown in studies of palaeofaeces (1000–2000 years old). This feces obtained in caves came from an era when obesity was nonexistent and shows the prominence of phylum Firmicutes before obesity became a problem [7]. Over the years, there has been a reduction in Firmicutes and increase in Bacteroidetes as the major phylum in the gut microbiota of western populations [7]. Increased obesity rates could be attributable to gut bacteria more efficient at extracting energy from food [8]. Knowledge of the role of specific bacterial species in the causation of the obese phenotype or resistance to becoming obese is minimal. More recent studies have shown that some genera among Firmicutes impact host obesity phenotype by increasing IgA secretion and reducing host lipid absorption through modulation proteins involved in the transport and absorption of fat in the small intestines, e.g. the expression of CD36 [9]. Variations in specific bacterial genera or species rather than phylum level differences may be more important.

To ascertain that the observation in outbred Sprague-Dawley CD rats [2,5] is also present in other species, we chose to study the microbiota composition of C57BL/6NTac mice because these mice are frequently used in obesity studies as a model of human obesity. When exposed to a HF diet, a small percentage does not rapidly gain weight and is excluded from obesity experiments. Besides being different species, the Sprague-Dawley CD rats and C57BL/6NTac mice are also reared in different environments (Charles River laboratory vs. Taconic Biosciences). We obtained 18–22-week-old mice that had been maintained on a HF diet (60% fat) since weaning. Twelve of the mice had responded to the diet and were obese, while 12 had retained a leaner frame despite exposure to the same diet. Because leptin production and leptin receptor sensing are not impaired in these mice, they do not display hormone-induced disturbances like those observed in ob/ob mice, db/db mice or the Zucker diabetic fatty rats. Herein, we sought to determine the association between the gut microbiota and intestinal and adipose tissue immune cell phenotype and the divergence in obesity phenotype. We report on the differences between the two phenotypes, particularly microbiota diversity, composition and function, and B and T cell phenotype in the lamina propria and adipose tissue.

## 2. Materials and Methods

### 2.1. Animals

All animal experiments and procedures were performed on 24 male C57BL/6NTac mice (Taconic Biosciences, Rensselear, NY, USA) in accordance with protocol number R-MAR-21–12 approved by the Institutional Animal Care and Use Committee (IACUC) of the University of Maryland, College Park (UMD). The mice used in this study were part of a larger colony that since weaning had been exposed to the same high-fat (HF) diet with 60% fat to induce obesity. From the larger colony, 12 mice that had responded to the HF diet and were obese (40–50 g) were randomly selected. From the same colony, 12 mice among those that retained a leaner frame and weighed significantly less (28–35 g) were randomly selected. All mice were aged 18–22 weeks. Once delivered to UMD, we categorized the mice that responded to diet-induced obesity (DIO) as ‘obese prone’ (OP) and categorized those that had not responded to DIO conditions as ‘obese resistant’ (OR). All mice were singly housed in controlled environmental conditions (22 °C) and 12-h light–dark cycles in shoebox cages containing corncob bedding. We maintained them on the same HF diet with 60% fat (Cat. # D12432; Research diets Inc. New Brunswick, NJ, USA) for 3 weeks with ad libitum access to food and water. Single housing was necessary to monitor the daily food/energy intake of each mouse, collect individual fecal samples of each mouse and reduce the cage effect in gut microbiome analysis.

### 2.2. Food Intake, Body Weight Insulin Resistance

Body weight and food intake for each mouse were monitored over 3 weeks, twice weekly. Blood for determination of fasting glucose and insulin was obtained by mandibular bleeding after fasting the animals for 6 h at the end of week 3. Glucose levels were determined by a portable glucometer and strips (Milpitas, CA, USA) and serum insulin levels were quantified by a mouse ELISA kit (Crystal Chem, Downers Grove, IL, USA). Insulin resistance was calculated as HOMA-IR = fasting glucose (mg/dL) × fasting insulin (ng/mL)/405 as shown before [10].

### 2.3. Euthanasia and Tissue Collection

After 4 weeks, mice were anesthetized by isoflurane inhalation for terminal blood collection by cardiac puncture and then euthanized by cervical dislocation. All required samples and tissues were harvested, weighed and snap-frozen in liquid nitrogen and later stored at −80 °C. The epididymal, perirenal, retroperitoneal and mesenteric fat pads were dissected, and their weight was summed as total abdominal fat.

### 2.4. Microbial DNA Extraction and Amplification

About 50 mg of small intestine contents (mixture from duodenum and jejunum) and fecal pellets were separately homogenized and lysed by bead beating using the FastPrep-24 (MP Biomedical, Solon, Ohio). DNA was extracted using the DNeasy PowerLyzer Power Soil Kit (Qiagen, Germantown, MD, USA) and quantified by the Qubit 4.0 Fluorometer (ThermoFisher Scientific, Rockville, MD, USA). From 1–3 ng of DNA, the 16S hypervariable regions V2-4-8 and V3-6, V7-9 were amplified using primers contained in the Ion 16S™ Metagenomics kit (ThermoFisher Scientific, Carlsbad, CA, USA). PCR conditions used were denaturation at 95 °C for 10 min, 25 cycles of denaturation at 95 °C for 30 s, annealing at 58 °C for 30 s, and extension at 72 °C for 20 s, with a final extension at 72 °C for 7 min on a CFX Connect™ Real-Time PCR system (BIO-RAD, Hercules, CA, USA). Amplification products from the two primer reactions were pooled, purified using Agencourt AMPure XP beads (Beckman Coulter, Nyon, Switzerland) on a DynaMag™-2 magnetic rack to remove primer dimers and small mispriming products, washed with fresh 70% ethanol, eluted into 15 ul of Nuclease-free water and quantified by the Qubit™ dsDNA HS Assay Kit (ThermoFisher Scientific, Carlsbad, CA, USA).

### 2.5. Library and Template Preparation and Sequencing

Libraries were prepared using end repair buffer and enzymes contained in the Ion Plus Fragment Library kit (ThermoFisher Scientific, Carlsbad, CA, USA) before purifying amplicons with AMPure XP beads on the DynaMag™-2 magnetic rack. Ion Xpress™ barcode adapters (ThermoFisher Scientific, Carlsbad, CA, USA) were used to ligate and barcode each sample followed by a thermal cycle run (25 °C for 15 min, 72 °C for 5 min) and cleanup before determining library concentration by Qubit™ dsDNA HS Assay kit. Exactly 10pM of the library was used for template preparation and enrichment using the Ion OneTouch™ 2 System. Template-positive Ion PGM™ Hi-Q™ Ion Sphere™ Particles (ISPs) with 400 base-pair average insert libraries were sequenced on an Ion 530™ Chip using the Ion GeneStudio™ S5 system (ThermoFisher Scientific, Grand Island, NY, USA).

### 2.6. Bioinformatics and Microbial Diversity Analysis

Demultiplexed and trimmed Fastq sequence files were curated from the Ion Reporter and processed using Quantitative Insights Into Microbial Ecology QIIME2-2022.2 as shown by Bolyen et al. [11]. Quality filtered demultiplexed sequences were denoised by removing chimeric sequences and correcting amplicon errors using the DADA2 plugin [12]. Amplicon sequence variants (ASVs) were determined and isolated. Forward and reverse reads were evaluated as single reads and then merged to produce full amplicons. Data were rarefied to a sampling depth of 10000 reads per sample and a phylogenetic tree was built with FastTree. Taxonomic classification was performed using the SILVA v138 database [13] and the MicroSEQ database v2013.1. We determined alpha diversity indices: observed ASVs (a count of unique ASVs in each sample), Pielou’s evenness (a measure of how close in numbers the species in a community are), Shannon diversity (to account for both abundance and evenness of the sequence variants present) and Faith-PD which uses phylogenetic distance (branch length of tree) to calculate the diversity in an individual sample. Beta diversity was calculated based on the weighted unifrac distance metric and visualized using principal component analysis (PCA) as implemented in QIIME2.

### 2.7. PICRUSt2

To predict the metabolic pathways, phylogenetic investigation of communities by reconstruction of unobserved states’ (PICRUSt2) pipeline was used as shown before [14]. Fastq sequence files were used as input files. Metabolic pathways were assigned based on the Kyoto Encyclopedia of Genes and Genomes (KEGG) Ortholog (KO) database. Read abundance data for all predicted pathways were converted to relative abundance and analyzed on the Galaxy server (https://huttenhower.sph.harvard.edu/galaxy/ (accessed: 30 April 2022)) for LEfSe using an LDA score of 3.0 as a threshold level to determine pathways most likely to explain differences between the OP and OR mice.

### 2.8. Isolation of Intestinal Cells for Flow Cytometry

We isolated cells in the lamina propria from the entire small intestine of four mice in each group in cold Roswell Park Memorial Institute (RPMI) buffer with 10% FBS after trimming off mesenteric fat and removing Peyers patches according to procedures described before [15,16]. Briefly, after cutting each sample in small segments, tissue was turned inside out, placed in extraction media (5% dithiothreitol, 0.5 M EDTA, 500 ul FBS in RPMI media), stirred at 500 rpm (28× *g*) for 15 min at 37 °C then removed while manually agitating to remove residual mucus. All segments were collected into a 1.5 mL tube containing digestion medium (25 mL RPMI, 2.5 mg dispase, 37.5 mg collagenase II and 300 ul FBS) followed by mincing and using a serological pipet to pipet up and down. Digested tissue was filtered through a 100 μm cell strainer into a 50 mL tube and the filtrate was centrifuged at 500× *g* for 10 min at 4 °C. The pellet was resuspended in 1 mL of flow cytometry staining buffer (ThermoFisher Scientific, Grand Island, NY, USA).

### 2.9. Staining and Flow Cytometry of Intestinal Immune Cells

We counted 1 × 10^6^ cells by Trypan Blue Exclusion Test and incubated them in FcR Blocking Reagent (Miltenyi Biotec Inc., Auburn, CA, USA) for 10 min at 4 °C. Dead cells were stained with 7-AAD Staining Solution (Miltenyi Biotec) for 5 min in the dark at room temperature. Cells were then stained for 30 min at 4 °C with fluorophore-conjugated monoclonal antibodies to CD3 (FITC), CD4 (VioBlue), B220 (VioGreen), CXCR5 (PE), PD-1 (APC), CD45 (FITC), CD19 (PE-Vio770), IgA (APC). Tregs were detected using a Treg detection kit (CD4/CD25/FoxP3) (Vio667) (Miltenyi Biotec Inc., Auburn, CA, USA). Flow cytometry was performed on a FACSCanto II and all data collected were analyzed using FlowJo™ v10.8 Software (BD Life Sciences, Franklin Lakes, NJ, USA).

### 2.10. Isolation of Adipose Tissue Cells for Flow Cytometry

Cells were isolated from the mesenteric adipose tissue (MAT), epididymal fat pad and peritoneal fat pad at euthanasia using the procedure descried in Orr et al. [17]. Briefly, 0.5 g tissue was washed and minced in 2 mL ice cold 1X DPBs (without Mg or Ca) supplemented with 0.5% BSA. The stromal vascular fraction (SVF) was isolated by collagenase II digestion (1X DPBS supplemented with 0.5% BSA, 10 mM CaCl2 and collagenase II). Erythrocytes were lysed by ACK lysis buffer. The remaining cell suspensions were filtered through a 70 μm strainer and resuspended in FACS buffer.

### 2.11. Staining and Flow Cytometry of Adipose Tissue Immune Cells

Approximately 1 × 10^6^ cells were incubated with FcR blocking reagents (Miltenyi Biotech) for 10 min before staining to block the unwanted binding of antibodies. Mouse flow cytometry antibodies: CD19, CD45, CD5 and IgM (Miltenyi Biotec) diluted to 1:50 in 100 μL FACS buffer were added to the cells. Samples were analyzed on a FACSCanto II. Data were analyzed by FlowJo™ Software v10.9. The IgM-producing B1b B cells were gated as ‘B220^mid/low^ CD19^high^CD5^−^IgM^+^’, with B220^mid/low^ CD^19high^ gated as B1 B cells, CD5^−^ is B-1b B cells (a subtype of B1 B cells) and IgM^+^ means total IgM production by B-1b B cells. The final data were represented as the percentage of the total population.

### 2.12. Determination of LPS and LPB in Serum

Because Lipopolysaccharide (LPS) is involved in obesity mechanisms, endotoxin levels were determined by quantifying serum LPS levels using a mouse sandwich ELISA kit (My Biosource, San Diego, CA, USA) and lipolysaccharide binding protein (LBP) Elisa kit (Abcam, Cambridge, UK) following the manufacturer’s provided protocols.

### 2.13. Analysis of Target Genes Encoding Proteins Involved in Fat Absorption, Macrophage Marker and Gut Barrier Integrity

We determined the relative expression of genes encoding proteins involved in fat absorption, macrophage infiltration in tissues, intestinal tight junctions and LPS sensing and signaling (TLR4) using validated primer sets (IDT Technologies, Coralville, IA, USA) listed in Appendix A. These proteins are all involved in different pathways linked to the pathogenesis of obesity. The RT-PCR cycling conditions on the CFX 96 (Bio-Rad, Hercules, CA, USA) were 2 min at 50 °C and 2 min at 95 °C, followed by 40 cycles of two-step PCR denaturation at 95 °C for 15 s and annealing extension at 60 °C for 1 min. Triplicate samples contained 10 ng cDNA and 6 μmol/L primers in 2× PowerUp™ SYBR™ Green Master Mix (Thermofisher Scientific, Rockville, MD, USA) in a final volume of 20 μL. The relative amount of target mRNA was normalized to Tata box protein (TPB) levels as an endogenous control gene. Data were analyzed by the 2^−ΔΔCT^ method, and the fold difference was calculated between the groups.

### 2.14. Statistical Analyses and Linear Discriminant Analysis

Data for food intake, body weight, abdominal fat pads, insulin resistance and flow cytometry and gene expression were compared by *t*-tests using GraphPad Prism 9. Data are expressed as mean ± SEM and significance was set at a two-tailed *p*-value < 0.05. The relative abundance of different bacterial taxa between OP and OR mice was converted to percentages and then analyzed by *t*-tests. Statistical significance for alpha and beta diversity was determined with Kruskal–Wallis and Permanova tests in QIIME2. Using the Galaxy online workflow application (https://huttenhower.sph.harvard.edu/galaxy (accessed on 30 April 2022)), we determined the taxa most likely to explain differences between the OP and OR mice using linear discriminant analysis (LDA) effect size (LEfSe). After converting data to log_10,_ we used a non-parametric Kruskal–Wallis rank sum test to assess differential features with significantly different abundances in assigned taxa and performed LDA to estimate the effect size of each sequence variant as reported by Segata et al. [18]. LDA scores ranking differential taxa were displayed on an LEfSe bar chart according to their effect size. A significant alpha level of 0.05 and an effect size threshold of 3 times greater difference was used for displaying results.

## 3. Results

### 3.1. Energy Intake, Body Weight, Body Fat and Insulin Resistance

All mice were fed the same diet (HF with 60% fat) and no differences in energy intake were observed (*p* = ns; Figure 1A). The OP mice exhibited a strong DIO responsiveness and gained more body weight, while the OR mice did not exhibit a DIO response (Figure 1B). The OR mice had significantly lower weights of perirenal and retroperitoneal fat pads (Figure 1D), lower liver weight (Figure 1E), lower fasting glucose, lower fasting insulin and subsequently higher insulin sensitivity (Figure 1F–G) (*p* < 0.0001) compared to OP mice. OP mice had a higher insulin resistance measure. Surprisingly, the epididymal fat pad weighed less in OP mice compared to OR mice (Figure 1C).

### 3.2. Rarefaction, Alpha and Beta Diversity Measures for Small Intestine Microbiota

The rarefaction curves for all samples reached a plateau, indicating that a moderate sequencing depth (ca. 10,000) was sufficient to capture the majority of the diversity in each group and detect the majority of ASVs in each sample (Figure 2). Rarefaction curves indicate that the small intestinal microbiota of OR mice had higher richness (more ASVs) compared to OP mice (Figure 2A–C) regardless of sequencing depth. Faith-PD, which takes phylogenetic distance into account and is confounded by richness, was also higher in OR mice (Figure 2C). Among the measures of alpha diversity, the observed ASVs value was not significantly different, but the Faith-PD index was significantly higher in OR mice compared to that of OP mice (*p* = 0.02). Evenness as measured by the Pielou-es index was not different between OP and OR mice (Figure 2D–F). The one-way Permanova non-parametric test of significant difference between groups, based on weighted unifrac distance, showed significant differences in bacterial composition (*pseudo-F* = 27.6; *p* < 0.025). Calculated distances within the OP group, within the OR group and between OP and OR groups are shown in Figure 2G while Figure 2H visualizes them in the PCA plot. Two distinct clusters defined by obesity phenotype were formed. No significant differences within group dispersion (permdisp) of bacteria community were detected between OP and OR mice (*pseudo-F* = 1.15; *p* = 0.52) (Figure 2I). Rarefaction, alpha and beta diversity measures for the lower gut (fecal samples) are shown in Appendix A.

### 3.3. Comparative Analysis of the Gut Microbiota Taxa Composition in Small Intestinal Contents

Relative abundances of bacterial taxa in the small intestines of OP vs. OR mice were calculated and compared graphically. The patterns observed in the alpha and beta diversity analysis were confirmed after taxonomic assignments as more taxonomic groups were observed in the intestines of OR mice (Figure 3 and Appendix A). More details are shown in Appendix A.

### 3.4. Firmicutes

The small intestine microbiota of OP mice was constituted almost entirely of Firmicutes at 98.2% while OR mice had 91.2% Firmicutes and Bacteroidota, Desulfobacterota and Proteobacteria detected at 3.6%, 3.7% and 0.7%, respectively (Figure 3A). Among classes, OP mice had higher amounts of Bacilli at 90.5% compared to 80.3% in OR mice. The OR mice had higher amounts of all other classes. Specifically, OR had Clostridia at 10.6% vs. 7.6% in OP mice, Desulfovibrionia 3.7 vs. 0.8% in OP mice, Bacteroidia at 3.6% vs. 0.7% in OP mice and Gammaproteobacteria at 0.6% vs. 0.03% in OR mice (Figure 3B). At order levels, Bacilli was composed almost entirely of order Lactobacillales, with OP mice having 88% while OR mice had 80%. Order Erysipelotrichales was found in much lower proportions with it constituting 2.5% in OP mice and only 0.04% in OR mice (Figure 3C). Among Clostridia, only order Peptostreptococcales-Tissierellales was present in higher proportions in OP mice at 5.8% vs. 2.6% in OR mice. Similarly, only the family Peptostreptococcaceae was higher in OP mice at 5.43% vs. 0.29% in OR mice. All other orders and families among Clostridia were more abundant in OR mice. Specifically, Clostridiales, Oscillospirales and Lachnospirales added up to about 8% in OR mice compared to 1.6% in OP mice (Figure 3C). Families Oscillospiraceae, *Lachnospiraceae* and Clostridiaceae added up to 7.9% in OR mice vs. 1.5% in OP mice. Among the order Lactobacillales, only two families Lactobacillaceae and Streptococcaceae were detected. In OP mice, the two families constituted 37.6% Lactobacillaceae and 50.3% Streptococcaceae. On the other hand, in OR mice, 82.6% of order Lactobacillales was constituted by Streptococcaceae and only 8.4% Lactobacillaceae (Figure 3C,D).

### 3.5. Bacteroidota

The phylum Bacteroidota consisted solely of the class Bacteroidia and order Bacteroidales and was represented by 0.73% in OP mice and 3.64% in OR mice. Each of the families under this phylum, i.e., *Muribaculaceae* (S24-7), Rikenellaceae, Marinifilaceae and Porphyromonadaceae, were present in higher proportions in OR mice compared to OP mice, with the three families adding up to 0.71 and 3.77% in OP and OR mice, respectively (Figure 3A–C).

### 3.6. Desulfobacterota

Phylum Desulfobacterota consisted solely of the class Desulfovibrionia, order Desulfovibrionales and family Desulfovibrionaceae at 3.89% in OR mice as compared to 0.78% in OP mice (Figure 3A–C).

### 3.7. Actinobacteria

The abundance of phylum Actinobacteria was very low in the small intestines and was not different between OP and OR mice, with just below 0.1% in either group.

### 3.8. Proteobacteria

Proteobacteria were present at very low amounts in both groups but were significantly higher in OR mice at 0.6% compared to 0.03% in OP mice. Only one class, the Gammaproteobacteria was detected (Figure 3A–C).

### 3.9. Linear Discriminant Analysis Effect Size (LEfSe)

The LEfSe algorithm was used to determine the effect size of significantly different abundances of taxa and visually rank them by the length of each bar. At a threshold of 3.0 on the logarithmic LDA score, the abundance of 22 taxa in the small intestines was different between the OP and OR mice and accounted for discriminative features between the mice (Figure 3E,F). A *p*-value less than 0.05 was considered significant in a non-parametric Kruskal–Wallis rank sum test. The LEfSe output was in agreement with abundances shown in the qualitative plots of taxa abundances (Figure 3A–D); family Streptococcaceae, specifically genus Lactococcus, was most abundant in OR mice while the family Lactobacillaceae, specifically genus Lactobacillus, was most abundant in OP mice, and these accounted most for differences between the two phenotypes.

### 3.10. Predicted Metabolic Functions

PICRUSt2 predicted 52 significantly different functional pathways by comparing them against KEGG orthologs. Data were transformed to relative abundance and differences between OP and OR mice are presented in Figure 4A. LEfSe analysis at a 3.0 threshold level and a *p*-value of 0.05 in a non-parametric Kruskal–Wallis rank sum test ranked the 30 pathways that accounted for discriminative features between the OP and OR mice (Figure 4B). The top five pathways identified as significantly higher in the microbiota of OR mice were the C5-branched dibasic metabolism; valine, leucine and isoleucine biosynthesis; phenylalanine, tyrosine and tryptophan biosynthesis; histidine metabolism; and Pantothenate Co-A metabolism. On the other hand, the pathways abundant in the microbiota of OP mice were the phosphotranferase system PTS, galactose metabolism, D-alanine metabolism, D-glutamine and D-glutamate metabolism, and fructose and mannose metabolism. A Pearson correlation analysis of major identified pathways against body fat and insulin resistance is shown in Figure 4C. The Pearson’s correlation coefficients are shown in each box. The C5-branched dibasic metabolism; valine, leucine and isoleucine biosynthesis; and phenylalanine, tyrosine and tryptophan biosynthesis pathways negatively correlated with body weight with an R^2^ of −0.47, −0.52 and −0.48, respectively. On the other hand, the phosphotranferase system (PTS), galactose metabolism and D-alanine metabolism pathways positively correlated with body weight with a Pearson correlation coefficient (r) of 0.59, 0.58 and 0.56, respectively.

### 3.11. Correlations between Body Weight, Insulin Resistance and Abundance in Specific Taxa and Alpha Diversity

Using Pearson correlation, coefficients between body weight and key bacterial taxa were identified, and also alpha diversity (Faith-PD). We did the same for insulin resistance. In the small intestines, the Faith-PD measure inversely correlated body weight (r = −0.36) and insulin resistance (r = −0.36). However, in the lower gut, the correlation was positive and much weaker with r = 0.25. Higher alpha diversity in the small intestines but not lower gut was thus associated with lower body weight and lower insulin resistance (Figure 5). We also performed correlations and taxa groups with differential abundance as identified by LefSE. Both body weight and insulin resistance significantly inversely correlated with bacterial taxa that were more abundant in OR mice. Genus *Lactococcus* and species *Lactococcus lactis* in particular had r = 0.64 and r = 0.63 in the lower gut. Family Desulfovibrionaceae and genus *Bilophila* in particular also had r = 0.35 in the small intestines and r= −0.32 in the lower gut. Family Lactobacillaceae had r = −0.20 in the small intestines and r = −0.27 in the lower gut. The families and genera identified as more abundant in OP mice all correlated positively with body weight and insulin resistance. Particularly, *Lactobacillus* had r = 0.72 and 0.43 for body weight and insulin resistance, respectively, in the small intestines. In the lower gut, the Pearson correlation coefficient between *Lactobacillus* was r = 0.66 and 0.70 for body weight and insulin resistance, respectively. In summary, a lean phenotype was associated with a higher alpha diversity and abundance of members of the genus *Lactococcus.* An obese phenotype was associated with a lower alpha diversity and abundance of members of the genus *Lactobacilli.*

### 3.12. Flow Cytometry of Small Intestinal Immune Cells

The population of CD3^+^ and CD4^+^ (T Helper cells) as a percentage of live lymphocytes trended higher in OP mice but was not significantly different from OR. The percentage of T_fh_ (CXCR5 PD1+) cells as a percentage of CD4^+^B220^−^ trended higher in OP mice but was not significantly different. However, specific subsets among these cells were different. The OP mice had a higher population of CD4^+^ Foxp3^+^ Tregs expressed as a percentage of CD4^+^ cells. Although the population of B cells (CD19^+^ as a percentage of CD4^+^ cells) was not different between the OP and OR mice, IgA^+^ B220^−^ cells were significantly more abundant in OR mice (Figure 6 and Appendix A).

### 3.13. Flow Cytometry of Adipose Tissue B Cells

The population of total B cells was significantly higher in the mesenteric fat pads of OR mice but there was no significant difference in the epididymal and perirenal fat pads. The subset population of B1b B IgM^+^ cells was significantly higher in the mesenteric and perirenal fat pads of OR mice. No significant differences were observed in the epididymal fat pad (Figure 7 and Appendix A).

### 3.14. LPS an LPB Expression in Serum

Results from the ELISA test for LPS in serum were inconclusive because our samples were degraded since we analyzed them a year after euthanasia. However, the amount of LPS binding protein (LBP) was significantly higher in OP mice (*p* < 0.05; Figure 8D).

### 3.15. Expression of Target Genes

The mRNA of CD36, the protein that carries fatty acids into cells, was significantly higher in OP mice in the small intestine, colon and adipose tissue (Figure 8A; *p* < 0.05). Expression of other genes encoding proteins involved in fat absorption and accumulation (chylomicron formation and triglyceride assembly or hydrolysis) like DGAT1, DGAT2, FATP4, Mogat1, Mttp and Mgll were not significantly different. Macrophage infiltration in tissues contributes to inflammation and eventually obesity. The expression of F4/80, the marker of macrophages, was significantly lower in the intestine, skeletal muscle and adipose tissue of OR mice compared to the OP (Figure 8B; *p* < 0.05).

The mRNA of claudin1 was significantly higher in OR mice (Figure 8C) but expression of Occludin, ZO-1 and MUC 2 were not different between OP and OR mice. The expression of TLR4, the sensing receptor for LPS and other antigens from bacteria in the colon, skeletal muscle and adipose tissue, was significantly lower in OR mice (Figure 8E; *p* < 0.05).

## 4. Discussion

Because all mice were fed the same diet, we sought to discern possible reasons for the different obesity phenotypes. While not a mechanistic study, it is a preliminary explanatory study to identify potential molecular targets for studies to promote resistance to obesity. In 18–22-week-old C57BL/6NTac mice classified as having responded to DIO or not, we showed that despite individual energy intake not being different, stark differences in the microbiota composition and function and immune cell phenotype in the lamina propria and adipose tissue are associated with the divergent obesity phenotype. Although a number of studies including Ridaura et al. [19] have demonstrated that microbial protection from adiposity is only possible against the backdrop of a suitable host diet, these mice exhibit a different phenotype although they are fed the same HF diet and share genetic background. Furthermore, mice are coprophagic (eat each other’s feces) and since weaning, these mice were group housed before being singly housed for three weeks only for purposes of determining individual energy intake and collection of individual fecal samples.

As expected, the OP mice which responded to the DIO regime accumulated more fat in the perirenal and peritoneal fat pads and in the liver and developed higher insulin resistance (Figure 1). Surprisingly, the epididymal fat pad of OP was smaller in size and weight compared to that of OR mice. However, reduced mass of the epididymal fat in mice with long-standing obesity has been observed by other research groups [20,21]. It is attributed to extensive cell death of adipocytes and inflammatory cell infiltration predominantly in the epididymal fat. Lipids released from dead adipocytes possibly find their way to the liver and other fat depots [20,21].

A high diversity and collective functional capacity of the gut microbiota is vital for optimal metabolic health at all stages of life. The small intestinal microbiota of OR mice had higher alpha diversity and evenness (Figure 2) and there was a distinct separate clustering between the microbiota of OR and OP mice (beta diversity) (Figure 2). Surprisingly, this was not the case in the lower gut, as alpha diversity was higher in OP mice and very weakly correlated with body weight or insulin resistance (Appendix A). Generally, persons whose microbiota has low gene richness (low diversity) are more likely to be obese [22,23]. We recently observed this same scenario in outbred Sprague-Dawley CD rats where a lower alpha diversity (richness and evenness) was observed in rats that responded to DIO while those that had a higher diversity remained leaner on the same diet [2]. In Sprague-Dawley CD rats, a greater abundance of Clostridia, particularly Clostridiaceae and *Lachnospiraceae*, accounted for most differences between obese prone and obese resistant rats. The mice in the current study were bred in a different environment, hence the predominant taxa were different. In agreement with the study in Sprague-Dawley CD rats [2], animals with enriched Bacteroidota, particularly family *Muribaculaceae*, had a higher probability of diet-induced obesity and insulin resistance (Figure 3 and Appendix A). *Muribaculaceae* are fermentative; carbohydrate-active enzymes constitute about 6% of the *Muribaculaceae* coding sequences, and based on enzyme abundance, the genes encode glycoside hydrolases, largely ɑ-amylases, suggesting starch as a key substrate with the ability to ferment or harvest energy from several carbohydrate moieties [22,23,24].

The outcome that stood out the most in our study was that differences in phenotype were associated with the proportions of *Lactococcus* and *Lactobacilli*. In OR mice, *Lactobacilli* constituted only 1.7% while *Lactococcus* was 38.4%. *Lactococcus* and *Lactobacilli* are both lactic acid producers widely used in processing fermented foods. It is clear that the metabolic outcomes that arise in these mice due to the abundance of these two genera are not attributed to the lactic acid. Although our study is not mechanistic, the observation that susceptibility to accrual of body fat correlates with proportions of *Lactococcus* and *Lactobacilli* (Figure 3 and Figure 5; Appendix A) is in agreement with experimental studies involving the provision of these two genera as probiotics against obesity [25,26]. Naudin et al. [25] compared supplementation of 1 × 10^9^ cfu *Lactococcus lactis* subsp cremoris and *Lactobacillus rhamnosus GG* for 16 weeks in mice on a HF diet regime. While *Lactobacillus rhamnosus GG* had no impact on body weight compared to mice given water as control, mice gavaged with *Lactococcus lactis* gained less body weight, gained less liver fat, had less inflammation, had reduced serum cholesterol and had better glucose tolerance compared with control mice. Furthermore, *Lactococcus lactis* lowered the respiratory exchanges ratio, increased night cycle activity, increased serum gastric inhibitory polypeptide (GIP) which stimulates insulin production, and lowered resistin, the adipose tissue hormone that is a biomarker of insulin resistance. *Lactobacillus rhamnosus* improved the augmentation of intestinal barrier function but not metabolic processes [26]. Another finding supporting the role of Lactococcus sp. in obesity prevention is by Zhang et al. [26] who showed that *Lactococcus chungangensis* CAU 28 alleviates diet-induced obesity and adipose tissue metabolism in vitro. In differentiated 3T3-L1 cells, *L. chungangensis* CAU 28 inhibited triglyceride formation and downregulated adipogenic transcription factors: fatty acid synthase, peroxisome proliferator-activated receptor-gamma and CCAAT-enhancer-binding protein-α, which are all associated with lipid accumulation. Furthermore, in a DIO mouse model, fed a HF diet, *L. chungangensis* CAU 28 attenuated body weight and abdominal and subcutaneous fat weight gain, increased serum adiponectin levels and decreased serum leptin levels [26]. The dominant taxa in each phenotype may have influenced fat absorption and metabolism. Gene expression of most proteins involved in fat absorption and metabolism was not different except for CD36, a protein that accelerates free fatty acid uptake, and extensive incorporation into triglycerides was more abundant in OP mice (Figure 8A).

PICRUSt2 analysis predicted that small intestinal bacteria in OR mice had an enrichment of genes encoding functions that synthesize amino acids valine, leucine, isoleucine, phenylalanine, tyrosine and tryptophan biosynthesis and C5-branched dibasic acid metabolism (Figure 4 and Appendix A). The C5-branched dibasic acid metabolism pathway involves five genes that catalyze the breakdown of amino acids and chemical reactions involving ATP and ADP. This finding is in agreement with a study in the Hadza people who are a traditional community with no exposure to western diets and/or obesity. Their gut microbiota was found to be highly enriched in genes responsible for branched-chain amino acid degradation and aromatic amino acid biosynthesis [6]. On the other hand, small intestinal bacteria of OP mice had an enrichment of bacterial genes encoding functions involved in carbohydrates metabolism, i.e., galactose metabolism, fructose and mannose metabolism and glycolysis and gluconeogenesis (Figure 4 and Appendix A). While the human genome encodes a set of enzymes capable of fully degrading a very small subset of glycans that have only one or two different linkages, i.e., starch, lactose and sucrose, bacterial genes encode enzymes capable of depolymerizing numerous glycans into their component monosaccharides. Gut bacteria vary widely in the number of different glycans that they can target [27]. It is possible that the microbiota of OP rats degrades more types of dietary glycans thus enhancing their capacity to harvest more energy from diet and hence promote more energy storage. While the OP and OR mice were on the same diets, the sensory inputs may differ in the intestinal environment and ultimately impact nutrient uptake and utilization by the bacteria [28,29,30]. Prediction by PICRUST identified bacterial chemotaxis as one of the pathways that correlates with body weight. The nutrient niches that may favor either *Lactococcus* or *Lactobacilli* are currently unknown. The host generates biogeographical microhabitats of nutrients through the release of antimicrobials or unknown resources that sculpt microbial growth by subdividing nutrient niches into biogeographical microhabitats for gut microbes [28,29,30]. We postulated that *Lactococcus* use motility and chemotaxis to reach a microhabitat in which their growth is fueled by favorable nutrients.

We sought to determine how the gut immune system is related to obesity phenotype. The intestinal lamina propria contains many types of myeloid and lymphoid cells that maintain tolerance or carry out inflammatory responses. Our results show that the population of CD3^+^ and CD4^+^ (T Helper cells) was not significantly different between the OP and OR mice. However, specific subsets among these cells were different. It was surprising that OP mice had a higher population of CD4^+^ Foxp3+ Tregs which promote the development of IgA-producing B cells in Peyer’s patches. Tregs are essential for maintaining a tolerant response to gut microbiota and are diminished in the presence of intestinal barrier abnormalities, and some studies have associated abundance of Tregs with a leaner phenotype [9]. However, our results were in conflict with OP mice showing higher population of Tregs. T cells also promote the development of IgA-producing B cells. IgA constrains the outgrowth of certain microbes and diversifies the microbiota [31]. Slight reductions in IgA negatively affect diversity and lead to systemic inflammation. The loss of IgA in mice worsens insulin resistance and glucose intolerance and increases intestinal microbiota encroachment and inflammation in metabolic tissues [31]. We found that although the population of B cells was not different between the OP and OR mice, IgA^+^ B cells were more abundant in OR mice. While a HF diet affects IgA-producing cells as well as sIgA levels, the significantly higher levels in OR mice likely contributed to the lower insulin resistance and better glucose tolerance. Persons with obesity and type 2 diabetes have lower mucosal IgA and pharmacological therapies for diabetes such as metformin and bariatric surgery increase cellular and stool IgA levels [32,33]. Tfh cells which play a critical role in helping B cells produce antibody were not different between OP and OR mice.

Besides fat accumulation, adipose tissue is a site of dietary-induced macrophage accumulation and inflammation contributing to systemic insulin resistance and glucose intolerance. B cells in adipose play a major role in orchestrating adipose tissue inflammation and insulin resistance. We thus determined differences between the B cell population in OP and OR mice. B1 cells, the major producers of IgM antibodies are prevalent in mucosal tissue and adipose tissue and provide first-line protection against bacterial pathogens and are important in obesity-related disorders [34,35]. The B1a and B1b subtypes of B1 cells possess some similarities in function but loss of B1b cells leads to inability to produce sufficient IgM and is associated with induction of insulin resistance. B1b cells are thought to reduce inflammation by impairing M1 macrophage infiltration, and IgM possesses an anti-inflammatory immune response that protects against metabolic dysfunction [36]. This finding may explain why the adipose tissue of OR had significantly lower levels of F4/80, the marker of macrophages (Figure 8B).

Because the LPS signaling system is related to pathogenesis of obesity, we compared serum LPS and LBP. The OP mice had higher amounts of LPB protein, which binds to LPS for transport, and higher expression of TLR4, the sensor of LPS in cells (Figure 8E) indicating that more inflammatory LPS is produced in these mice. Even if LPS production was not different, the OR mice were maybe protected against LPS because their gut barrier had higher integrity (Figure 8D).

## 5. Conclusions

Our findings support associations between intestinal microbiota function with propensity to diet-induced obesity. Particularly, that microbiota of the small intestine correlates higher to diet-induced obesity compared to lower gut microbiota. In addition, immune cell phenotype in the intestine and adipose may contribute to the different phenotypes. The limitation of our study is that it is not mechanistic. However, this preliminary study lays a foundation of mechanistic studies on how particular components of the gut microbiota or immune system in intestine or adipose can be targeted to promote resistance to obesity.

## Figures and Tables

**Figure 1 microorganisms-11-02153-f001:**
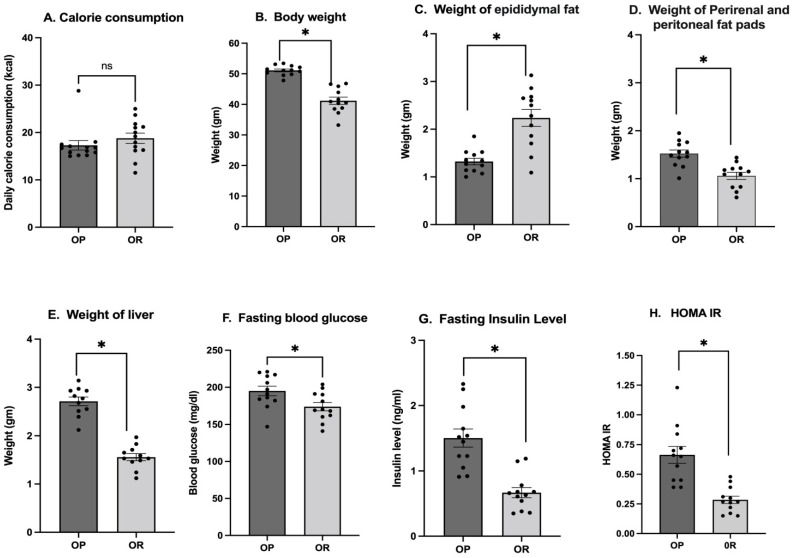
**Despite similar amounts of daily calorie intake, OR mice retained a lower body weight, less fat accumulation and lower insulin resistance compared to OP mice.** Body weight, liver and adipose tissue weight were determined at euthanasia. Fasting glucose and fasting insulin were determined by glucometer and ELISA test, respectively, after 6 h of fasting. (**A**) Weekly calorie intake was not different (*p* = ns). (**B**) Body weight was significantly higher among OP mice compared to OR mice (*p* < 0.001). (**C**) The weight of the epididymal fat pad was significantly lower in OP mice compared to OR mice (*p* < 0.001). (**D**) The perirenal and peritoneal fat pads weighed significantly higher in OP mice compared to OR mice (*p* < 0.001). (**E**) Liver weight was significantly higher in OP mice compared to OR mice (*p* < 0.001). (**F**) Fasting glucose was significantly higher in OP mice compared to OR mice (*p* = 0.02). (**G**) Fasting insulin levels were significantly higher in OP mice compared to OR mice (*p* < 0.0001). (**H**). The insulin resistance index HOMA-IR was significantly higher in OP mice compared to OR (*p* < 0.001). (gm = grams). * *p* < 0.05; *n* = 12.

**Figure 2 microorganisms-11-02153-f002:**
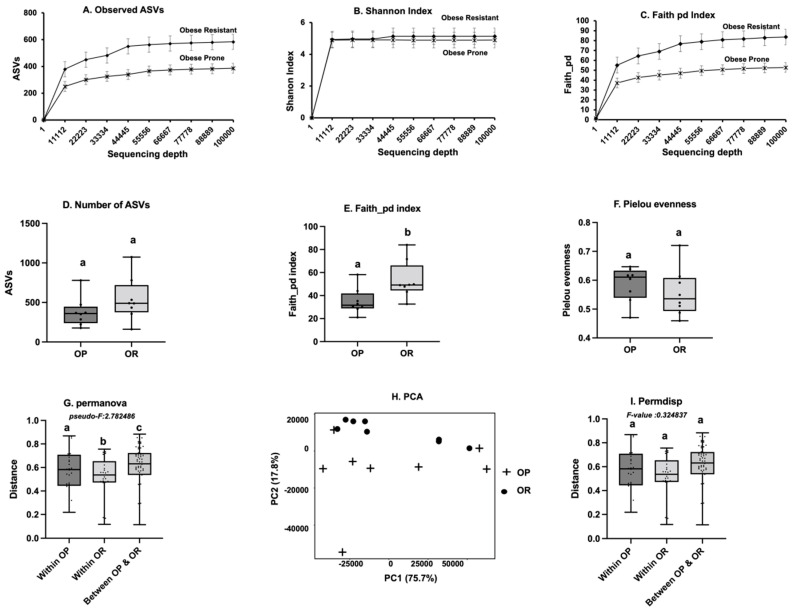
**Diversity measures among small intestinal contents of OR and OP mice.** (**A**–**C**). Observed ASVs and Faith-PD were both higher in OR mice (*p* < 0.001). The Shannon index was not different between OP and OR (*p* = ns). (**D**–**F**). The alpha diversity index as determined by Faith-PD was higher in OR mice (*p* = 0.02, *n* = 12) but the observed ASVs were not different (*p* = ns). The Pielou-e evenness index was not different among OR or OP (*p* = ns). (**G**). The one-way Permanova non-parametric test of significant difference between groups, based on weighted unifrac distance, showed significant differences between OP and OR bacteria composition (*pseudo-F* = 27.6; *p* = 0.025). The first box plot is the intra group variability within OP, the second box plot is the intra group variability within OR and the third box plot is the variability between OP and OR groups. (**H**). The visualization of the weighted unifrac distance between the OP and OR mice. (**I**). Weighted unifrac Permdisp (dispersion) in the bacterial communities was not different between OP and OR mice (*p* = ns). ‘a–c’ Diffrenet letters indicate significant difference at *p* < 0.05.

**Figure 3 microorganisms-11-02153-f003:**
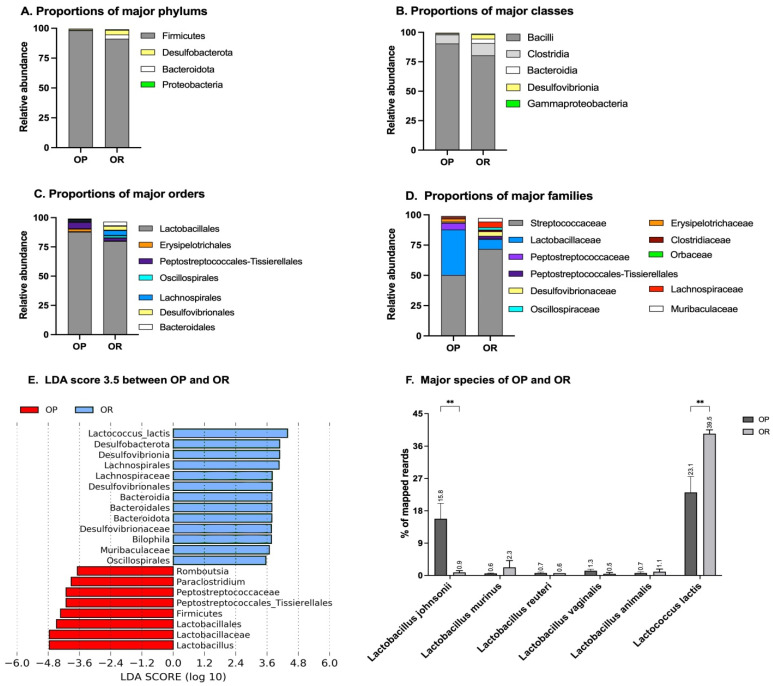
**The relative abundance of bacterial groups in small intestines.** 16S sequencing data showed that proportions of taxa between the OP and OR phenotypes were different. (**A**). While only Firmicutes was prominent in OP mice, OR mice had more quantifiable phyla observed: Firmicutes, Desulfobacterota, Bacteroidota and Proteobacteria. (**B**). OR mice had lower amounts of class Bacilli and higher amounts of Clostridia, Bacteroidia and Desulvobrionia compared to OP mice. The OP mice mainly had only Clostridia and Bacilli detected. (**C**). OR mice had higher amounts of orders Clostridiales, Bacteroidales, Desulfovibrionales, Oscillospirales and Lachnospirales compared to OP mice. The OP mice had higher Lactobacillales. (**D**). OR mice had higher amounts of families Streptococcaceae, *Lachnospiraceae*, Clostridiaceae, Desulfovibrionaceae, Oscillospiraceae and *Muribaculaceae* while OP had higher Lactobacillaceae. (**E**). Differential abundance of taxa was ranked according to their effect size, and discriminative taxa were selected based on an LDA score cutoff of 3.0. Differences in the relative abundances of taxa (converted to log base 10) were statistically determined based on Kruskal–Wallis and pairwise Wilcoxon tests at *p* < 0.05. The length of the histogram represents the LDA score, i.e., the degree of influence of species with significant differences between different groups. (**F**). Proportions of Lactococcus and Lactobacillus species identified in OP and OR mice. Data are means ± SEM. ** *p* < 0.01.

**Figure 4 microorganisms-11-02153-f004:**
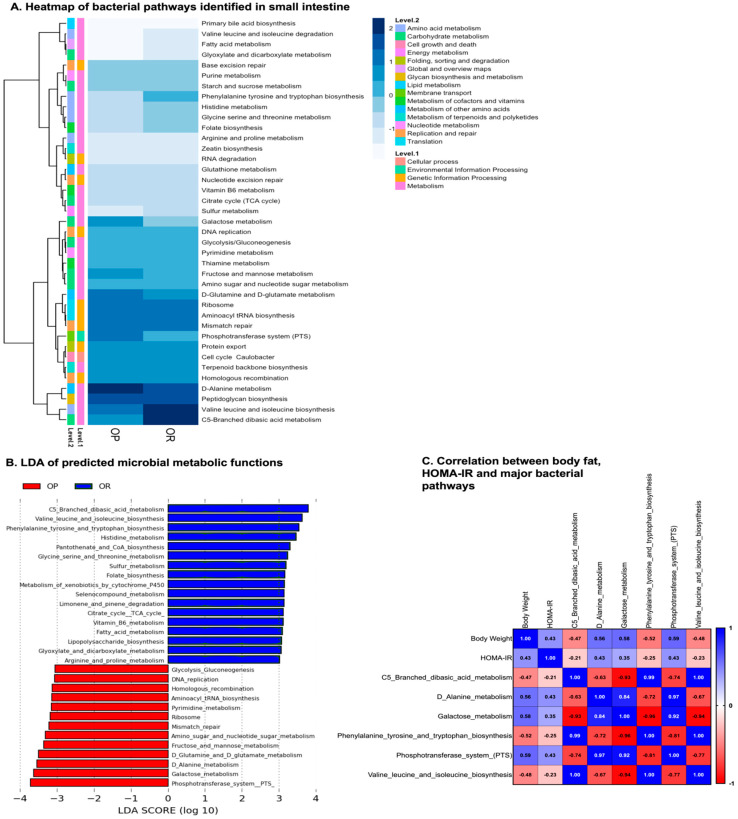
**Comparison of KEGG pathways predicted by PICRUSt**. Metabolic pathways by PICRUSt2 using 16S rRNA gene sequences were predicted from the Kyoto Encyclopedia of Genes and Genomes (KEGG) Ortholog (KO) database. (**A**). Heatmap of the 40 s-level classification KEGG pathways identified in the intestinal contents. (**B**). LDA scores calculated for the differentially abundant features in the small intestinal bacteria. Selection of discriminative microbial pathways between groups was based on an LDA score cutoff of 3.0 and differences in the relative abundances of the pathway (converted to log base 10) were statistically determined based on a Kruskal–Wallis and pairwise Wilcoxon tests. A *p*-value of <0.05 and a score ≥3.0 were considered significant in Kruskal–Wallis and pairwise Wilcoxon tests, respectively, at a significance level of 0.05; *n* = 12. The length of the histogram represents the LDA score, i.e., the degree of influence of species with significant difference between different groups. (**C**). Correlation analysis of body fat and insulin resistance with major microbial pathways identified. The R^2^ with *p* < 0.05 value is shown in each cell.

**Figure 5 microorganisms-11-02153-f005:**
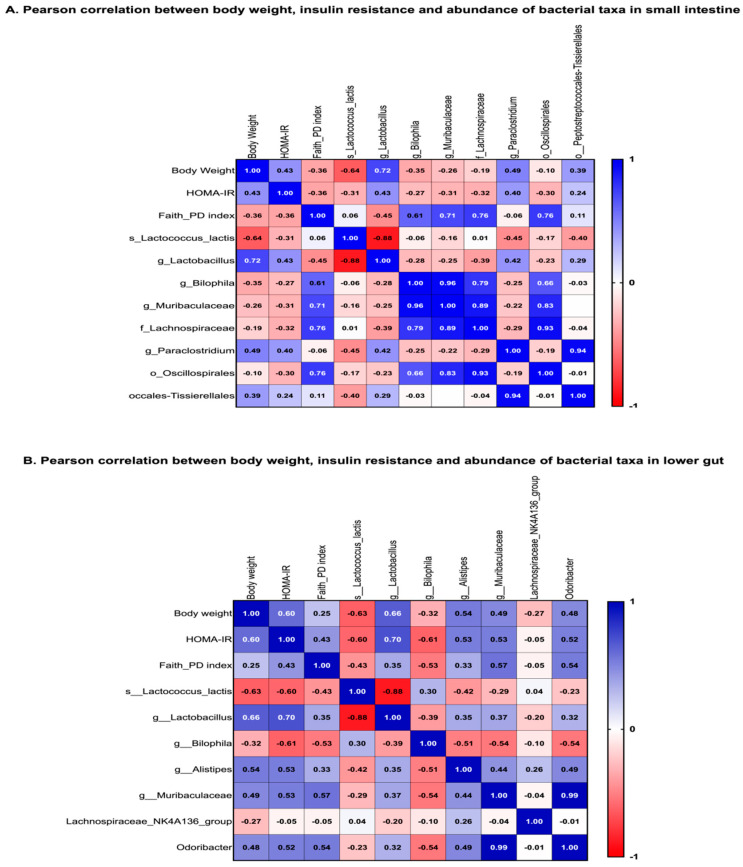
**Pearson correlation between body weight, insulin resistance and abundance of bacterial taxa in the small intestines and lower gut (fecal contents).** (**A**) In the small intestines, α diversity and abundance of *Lactococcus*, *Bilophila*, and *Lachnospiraceae* all inversely correlated with body weight and insulin resistance. The Pearson correlation coefficient ‘r’ at a significance of *p* < 0.05 is shown in each box. The highest r value was for *Lactococcus lactis* at −0.64 and −0.31 for body weight and insulin resistance, respectively. Abundance of Lactobacilli positively correlated with body weight and insulin resistance with r of 0.72 and 0.43, respectively. (**B**) In the lower gut, *Lactococcus*, *Bilophila* and *Lachnospiraceae* all inversely correlated with body weight and insulin resistance. The highest r value was for *Lactococcus lactis* at r = −0.63 and −0.60 for body weight and insulin resistance, respectively. Alpha diversity positively correlated with body weight and insulin resistance with r = 0.25. *Lactobacilli* positively correlated with body weight and insulin resistance with r = 0.66 and 0.7, respectively. *Muribaculaceae* positively correlated with body weight and insulin resistance with r = 0.49 and 0.53, respectively.

**Figure 6 microorganisms-11-02153-f006:**
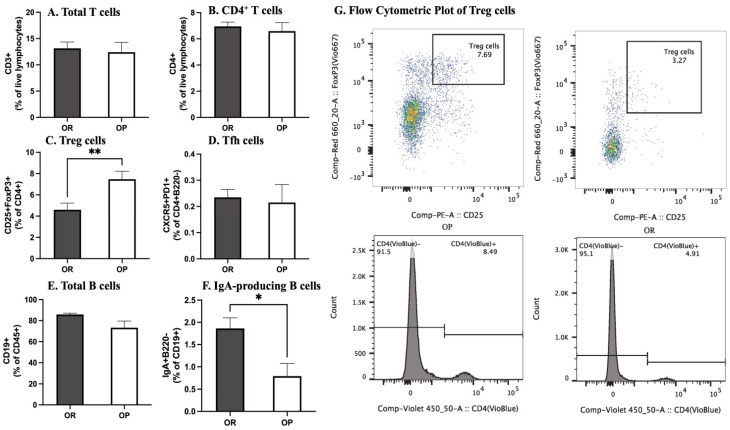
**Flow Cytometry of B and T cells in the Lamina Propria.** *Lamina propria* (*n* = 12/group, 4 experiments) in HFD-fed OP and OR mice. (**A**,**B**). Frequency of CD3^+^, CD4^+^. (**C**). Frequency of CD4^+^ CD25^+^ Foxp3^+^ Tregs. (**D**). CXCR5 PD1^+^ T cells as a ratio of CD4^+^ B220^−^ cells. (**E**). Frequency of CD19^+^ cells. (**F**). Frequency of IgA^+^ B220^−^ cells as a percentage of CD19^+^. (**G**). Representative raw data for quantification of Tregs. Data are means ± SEM. * denotes *p* < 0.05, ** denotes *p* < 0.01.

**Figure 7 microorganisms-11-02153-f007:**
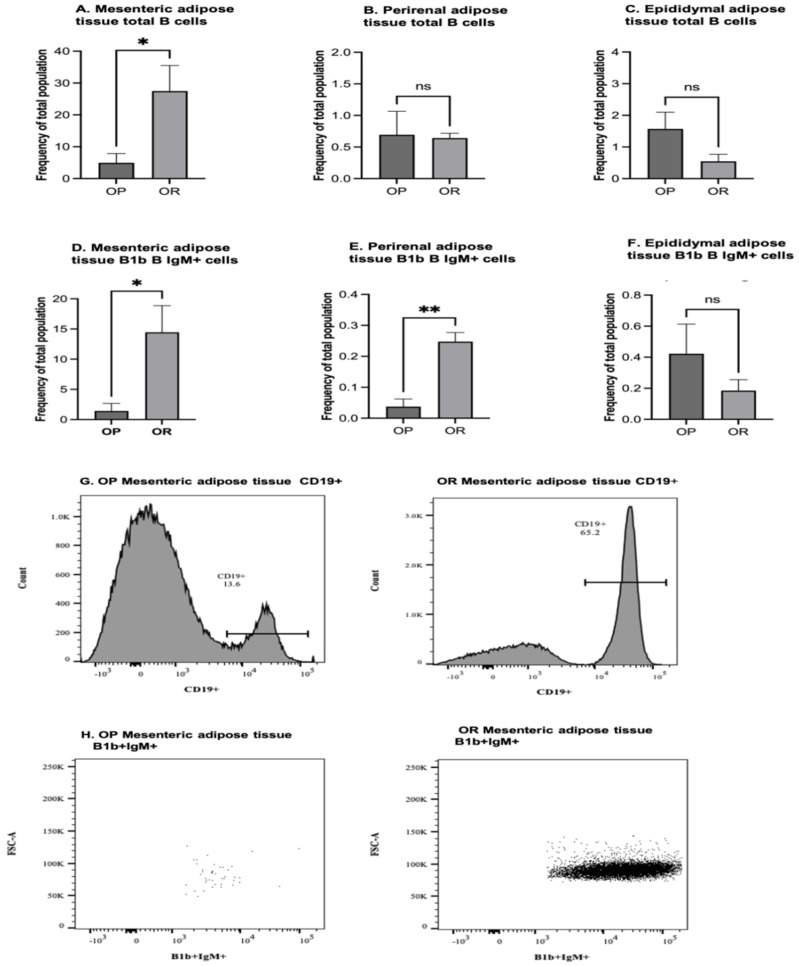
**Flow Cytometry of B cells in Adipose tissue fat pads.** Adipose tissue fat pads (*n* = 12/group, 4 experiments) in HFD-fed OP and OR C57BL/6J mice. (**A**–**C**). Frequency of CD19^+^ cells in mesenteric, perirenal and epididymal fat pads, respectively. (**D**–**F**). Frequency of B1b B IgM^+^ cells in mesenteric, perirenal and epididymal fat pads, respectively. (**G**,**H**). Representative raw data for quantification of CD19^+^ cells and B1b B IgM^+^ cells in adipose tissue. Data are means ± SEM. * denotes *p* < 0.05, ** denotes *p* < 0.01, ns = not significant.

**Figure 8 microorganisms-11-02153-f008:**
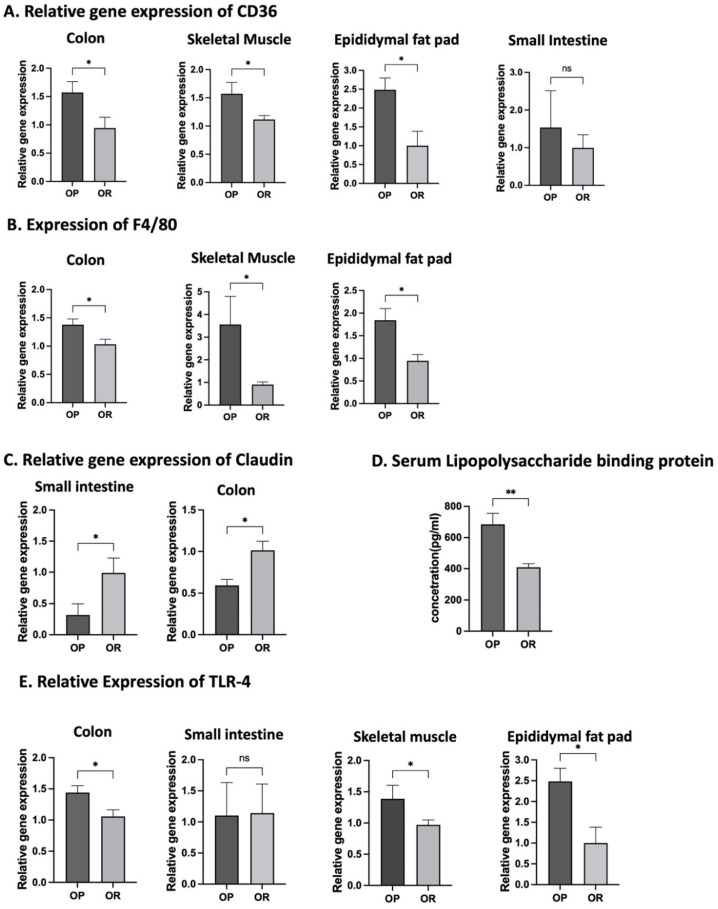
**Gene expression and protein determination of obesity markers.** Gene expression of CD36, F4/80, Claudin and TLR4 was determined by RT-qPCR from RNA extracted from respective tissues. Serum LPB was determined by ELISA kit. (**A**). mRNA expression of CD36 in the colon, small intestine, skeletal muscle and adipose tissue. (**B**). mRNA expression of F4/80 in the colon, skeletal muscle and adipose tissue. (**C**). m-RNA expression of the tight junction protein Claudin in intestinal tissue and colon. (**D**). Quantities of LBP in serum. (**E**). mRNA expression of TLR4 in colon, small intestine, skeletal muscle and adipose tissue. Data are means ± SEM. * denotes *p* < 0.05, ** denotes *p* < 0.01, ns = not significant.

## Data Availability

All data generated is available in the manuscript and Appendix A.

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
