# Peer review of "Resistance to Diet Induced Visceral Fat Accumulation in C57BL/6NTac Mice Is Associated with an Enriched Lactococcus in the Gut Microbiota and the Phenotype of Immune B Cells in Intestine and Adipose Tissue"

_microorganisms, 2023, doi:10.3390/microorganisms11092153_

Round 1
Reviewer 1 Report
Check the spelling and error
1: Please make a short title.
2: Highlight is so many look like an abstract (make short)
3: re-write the abstract with a clear objective.
4: Enrich introduction with references.
Author Response
Comments from Reviewer 1
Comment 1: Please make a short title.
Response: We have done our best to revise the title and improve clarity but to capture the findings of the study we are unable to eliminate using two ‘and’s in the sentence.
Comment 2: Highlight is so many look like an abstract (make short)
Response: Thank you for pointing this out. The highlight points have been merged into four key points.
Comment 3: re-write the abstract with a clear objective.
Response: We have re-written the hypothesis to read: We hypothesized that in C57BL/6NTac mice, despite a shared genetic background and diet, variations in individual gut microbiota function, immune cell phenotype in the intestine and adipose determine predisposition to obesity.
Comment 4: Enrich the introduction with references.
Response: We have added two more referances and re-written the introduction. The referances added are below:
- Stephens RW and Arhire L. Gut Microbiota: From microorganisms to metabolic organ influencing obesity. Obesity26(5):801-809. Wiley 2018. 1930-7381. 10.1002/oby.22179
- Boekhorst, J., Venlet, N., Procházková, N. et al.Stool energy density is positively correlated to intestinal transit time and related to microbial enterotypes. Microbiome 10, 223 (2022). https://doi.org/10.1186/s40168-022-01418-5.
Reviewer 2 Report
Title: “Resistance to diet induced visceral fat accumulation in C57BL/6NTac mice is associated with an enriched Lactococcus in the gut microbiota and intestinal and adipose tissue immune cell phenotype.
In addition to the substantive changes and clarifications made as track changes, it is important to address the following technical errors present in the manuscript.
1. Firstly, there are several typographical errors throughout the manuscript that should be addressed. Please review the text thoroughly and correct any identified typos.
2. Secondly, inconsistencies in formatting are present in various sections of the manuscript. It is important to ensure consistent and appropriate formatting to enhance readability and maintain a professional presentation.
3. Additionally, I noticed an excessive use of italics and highlighting. It is recommended to use these formatting options sparingly and only when necessary for emphasis or to convey specific information.
4. Furthermore, several grammatical mistakes have been identified. It is crucial to carefully proofread the text, paying close attention to sentence structure, verb tense consistency, and subject-verb agreement.
5. it is crucial to highlight that the English language usage within the manuscript necessitates thorough proofreading and improvement.
I appreciate the effort put into this work, and I believe that addressing these comments will significantly enhance the quality of the manuscript. For detailed track changes and additional comments, please refer to the attached file.

English language used requires proofreading
Author Response
Comments from Reviewer 2
Comment 1: Firstly, there are several typographical errors throughout the manuscript that should be addressed. Please review the text thoroughly and correct any identified typos.
Response: We have checked for typos and corrected and edited. We have accepted most of the suggested changes.
Comment 2: Secondly, inconsistencies in formatting are present in various sections of the manuscript. It is important to ensure consistent and appropriate formatting to enhance readability and maintain a professional presentation.
Response: We have checked for formatting issues and corrected them after editing. We have accepted most of the suggested changes
Comment 3: Additionally, I noticed an excessive use of italics and highlighting. It is recommended to use these formatting options sparingly and only when necessary for emphasis or to convey specific information.
Response: We have checked for formatting issues and corrected them after editing. We have accepted most of the suggested changes
Comment 4: Furthermore, several grammatical mistakes have been identified. It is crucial to carefully proofread the text, paying close attention to sentence structure, verb tense consistency, and subject-verb agreement.
Response: We have checked for grammatical errors and corrected them after editing. We have accepted most of the suggested changes
Comment 5: it is crucial to highlight that the English language usage within the manuscript necessitates thorough proofreading and improvement.
Response: The grammatical errors have been identified and corrected. The English proofreading was also performed to enhance comprehensibility.
Apart from the comment, a Word document addressing reviewer 2’s comments has been uploaded.

Reviewer 3 Report
This paper aimed to investigate resistance to diet induced visceral fat accumulation in C57BL/6NTac mice is associated with an enriched Lactococcus in the gut microbiota and intestinal and adipose tissue immune cell phenotype, which was of general interest to Nutrients. However, the paper should be considerably improved. I have listed some comments below:
1. Nutrients seem to not require Highlights. If necessary, please reduce the Highlights to 4-5 entries.
2. Please do not divide the Abstract section into two paragraphs.
3. Please provide the conditions for experimental animals, such as environmental temperature, humidity, light, etc.
4. Why the author's 16S sequencing be divided into V2-4-8 and V3-6, 7-9 detection segments. There are repeated intervals in these paragraphs. How did the author merge the data.
5. Please convert all rpm to rcf or g.
6. Figure 1B and C. What does gm mean? Is it g per mouse? If so, simply write it as g.
7. Why does the first paragraph of results need to be italicized? Does it have any special meaning?
8. There are problems with gene analysis and data processing, which must be corrected. If calculated based on 2−ΔΔCT, the relative gene expression level of one of the groups in the results should be 1.
9. There should be spaces between units and numbers.
10. Please unify the “Fig.” or “Figure” in the manuscript.
11. Please unify the form of P value, p or P.
12. Please italicize the name of the genus level of the bacteria.
13. Figure 3F. The results of 16S sequencing can only be accurate to the genus level. How are species level data derived?
14. Figure 6. Please provide an isotype control for CD4+CD25+FoxP3+ and scatter plots for the rest of the antibody markers.
15. Figure 8. The marker of obesity should be TC, TG, HDL, LDL. Rather than the indicators presented by the authors. Why did the author not detect TC, TG, HDL, LDL? It is suggested to supplement relevant data.
16. The format of the References should be checked (refer to Authors Guidelines).
Moderate editing of English language required.
Author Response
Comment 1: Nutrients seem to not require Highlights. If necessary, please reduce the Highlights to 4-5 entries.
Response: Thank you for pointing this out. The highlight points have been merged into four key points.
Comment 2: Please do not divide the Abstract section into two paragraphs.
Response: The abstract has been modified to one paragraph.
Comment 3: Please provide the conditions for experimental animals, such as environmental temperature, humidity, light, etc.
Response: This information has been updated in the method section as “All mice were singly housed in controlled environmental conditions (22°C), 12-hour light-dark cycle in shoebox cages containing corncob bedding”
Comment 4: Why the author's 16S sequencing be divided into V2-4-8 and V3-6, 7-9 detection segments? There are repeated intervals in these paragraphs. How did the author merge the data?
Response: The 16S amplification kit of the manufacturers (Thermofisher Scientific) comes with two primers targeting of V2-4-8 and V3-6, 7-9 regions. The 16S amplification was done by PCR as per the manufacturer’s instructions to prepare the final library. The Ion Torrent platform generates single-end fastq files per sample.
Comment 5: Please convert all rpm to rcf or g.
Response 5: Done. Only one place had a centrifuge speed identified
Comment 6: Figure 1B and C. What does gm mean? Is it g per mouse? If so, simply write it as g.
Response 6: gm of Figure 1B and C is the abbreviation for gram. The information has been updated in the legends of Figure 1.
Comment 7: Why does the first paragraph of results need to be italicized? Does it have any special meaning?
Response 7: Thank you for pointing this out. The formatting issue has been corrected.
Comment 8: There are problems with gene analysis and data processing, which must be corrected. If calculated based on 2−ΔΔCT, the relative gene expression level of one of the groups in the results should be 1.
Response 8: The gene expression was recalculated, and Figure 9 was updated with new figures generated from the reanalysis of the gene expression by the 2−ΔΔCT method.
Comment 9: There should be spaces between units and numbers.
Response 9: All edits are done.
Comment 10: Please unify the “Fig.” or “Figure” in the manuscript.
Response 9: Done. All converted to Figure.
Comment 11: Please unify the form of P value, p, or P.
Response:11: Formatting issues have been addressed. All p are now P.
Comment 12: Please italicize the name of the genus level of the bacteria.
Response 12: Formatting issues have been addressed.
Comment 13: Figure 3F. The results of 16S sequencing can only be accurate to the genus level. How are species-level data derived?
Response 13: The sequencing data at the species level was identified from the MicroSEQ database v2013.1 a comprehensive resource for quality checked and aligned ribosomal RNA sequence data that is proprietary to Thermofisher Scientific for use after sequencing on the Ion Torrent platform. MicroSEQ identifies more species compared to SILVA.
Comment 14: Figure 6. Please provide an isotype control for CD4+CD25+FoxP3+ and scatter plots for the rest of the antibody markers.
Response 14: These are now shown as Supplementary Figures 4-5.
Comment 15: Figure 8. The marker of obesity should be TC, TG, HDL, LDL. Rather than the indicators presented by the authors. Why did the author not detect TC, TG, HDL, LDL? It is suggested to supplement relevant data.
TC, TG, HDL, LDL are markers of cardiovascular health, insulin resistance and are good readouts for an obesity study. However they will all increase in response to a high fat diet which all animals in this study were exposed to. Our focus was on possible causes of the different fat accumulation phenotype in adipose tissue. Secondly the serum samples are more than 2 years old and may be unsuitable for analysis.
Comment 16: The format of the References should be checked (refer to Authors Guidelines).
All referances have been rechecked and formatted to be the same format.
Comment 1: Nutrients seem to not require Highlights. If necessary, please reduce the Highlights to 4-5 entries.
Response: Thank you for pointing this out. The highlight points have been merged into four key points.
Comment 2: Please do not divide the Abstract section into two paragraphs.
Response: The abstract has been modified to one paragraph.
Comment 3: Please provide the conditions for experimental animals, such as environmental temperature, humidity, light, etc.
Response: This information has been updated in the method section as “All mice were singly housed in controlled environmental conditions (22°C), 12-hour light-dark cycle in shoebox cages containing corncob bedding”
Comment 4: Why the author's 16S sequencing be divided into V2-4-8 and V3-6, 7-9 detection segments? There are repeated intervals in these paragraphs. How did the author merge the data?
Response: The 16S amplification kit of the manufacturers (Thermofisher Scientific) comes with two primers targeting of V2-4-8 and V3-6, 7-9 regions. The 16S amplification was done by PCR as per the manufacturer’s instructions to prepare the final library. The Ion Torrent platform generates single-end fastq files per sample.
Comment 5: Please convert all rpm to rcf or g.
Response 5: Done. Only one place had a centrifuge speed identified
Comment 6: Figure 1B and C. What does gm mean? Is it g per mouse? If so, simply write it as g.
Response 6: gm of Figure 1B and C is the abbreviation for gram. The information has been updated in the legends of Figure 1.
Comment 7: Why does the first paragraph of results need to be italicized? Does it have any special meaning?
Response 7: Thank you for pointing this out. The formatting issue has been corrected.
Comment 8: There are problems with gene analysis and data processing, which must be corrected. If calculated based on 2−ΔΔCT, the relative gene expression level of one of the groups in the results should be 1.
Response 8: The gene expression was recalculated, and Figure 9 was updated with new figures generated from the reanalysis of the gene expression by the 2−ΔΔCT method.
Comment 9: There should be spaces between units and numbers.
Response 9: All edits are done.
Comment 10: Please unify the “Fig.” or “Figure” in the manuscript.
Response 9: Done. All converted to Figure.
Comment 11: Please unify the form of P value, p, or P.
Response:11: Formatting issues have been addressed. All p are now P.
Comment 12: Please italicize the name of the genus level of the bacteria.
Response 12: Formatting issues have been addressed.
Comment 13: Figure 3F. The results of 16S sequencing can only be accurate to the genus level. How are species-level data derived?
Response 13: The sequencing data at the species level was identified from the MicroSEQ database v2013.1 a comprehensive resource for quality checked and aligned ribosomal RNA sequence data that is proprietary to Thermofisher Scientific for use after sequencing on the Ion Torrent platform. MicroSEQ identifies more species compared to SILVA.
Comment 14: Figure 6. Please provide an isotype control for CD4+CD25+FoxP3+ and scatter plots for the rest of the antibody markers.
Response 14: These are now shown as Supplementary Figures 4-5.
Comment 15: Figure 8. The marker of obesity should be TC, TG, HDL, LDL. Rather than the indicators presented by the authors. Why did the author not detect TC, TG, HDL, LDL? It is suggested to supplement relevant data.
TC, TG, HDL, LDL are markers of cardiovascular health, insulin resistance and are good readouts for an obesity study. However they will all increase in response to a high fat diet which all animals in this study were exposed to. Our focus was on possible causes of the different fat accumulation phenotype in adipose tissue. Secondly the serum samples are more than 2 years old and may be unsuitable for analysis.
Comment 16: The format of the References should be checked (refer to Authors Guidelines).
All referances have been rechecked and formatted to be the same format.
Comment 1: Nutrients seem to not require Highlights. If necessary, please reduce the Highlights to 4-5 entries.
Response: Thank you for pointing this out. The highlight points have been merged into four key points.
Comment 2: Please do not divide the Abstract section into two paragraphs.
Response: The abstract has been modified to one paragraph.
Comment 3: Please provide the conditions for experimental animals, such as environmental temperature, humidity, light, etc.
Response: This information has been updated in the method section as “All mice were singly housed in controlled environmental conditions (22°C), 12-hour light-dark cycle in shoebox cages containing corncob bedding”
Comment 4: Why the author's 16S sequencing be divided into V2-4-8 and V3-6, 7-9 detection segments? There are repeated intervals in these paragraphs. How did the author merge the data?
Response: The 16S amplification kit of the manufacturers (Thermofisher Scientific) comes with two primers targeting of V2-4-8 and V3-6, 7-9 regions. The 16S amplification was done by PCR as per the manufacturer’s instructions to prepare the final library. The Ion Torrent platform generates single-end fastq files per sample.
Comment 5: Please convert all rpm to rcf or g.
Response 5: Done. Only one place had a centrifuge speed identified
Comment 6: Figure 1B and C. What does gm mean? Is it g per mouse? If so, simply write it as g.
Response 6: gm of Figure 1B and C is the abbreviation for gram. The information has been updated in the legends of Figure 1.
Comment 7: Why does the first paragraph of results need to be italicized? Does it have any special meaning?
Response 7: Thank you for pointing this out. The formatting issue has been corrected.
Comment 8: There are problems with gene analysis and data processing, which must be corrected. If calculated based on 2−ΔΔCT, the relative gene expression level of one of the groups in the results should be 1.
Response 8: The gene expression was recalculated, and Figure 9 was updated with new figures generated from the reanalysis of the gene expression by the 2−ΔΔCT method.
Comment 9: There should be spaces between units and numbers.
Response 9: All edits are done.
Comment 10: Please unify the “Fig.” or “Figure” in the manuscript.
Response 9: Done. All converted to Figure.
Comment 11: Please unify the form of P value, p, or P.
Response:11: Formatting issues have been addressed. All p are now P.
Comment 12: Please italicize the name of the genus level of the bacteria.
Response 12: Formatting issues have been addressed.
Comment 13: Figure 3F. The results of 16S sequencing can only be accurate to the genus level. How are species-level data derived?
Response 13: The sequencing data at the species level was identified from the MicroSEQ database v2013.1 a comprehensive resource for quality checked and aligned ribosomal RNA sequence data that is proprietary to Thermofisher Scientific for use after sequencing on the Ion Torrent platform. MicroSEQ identifies more species compared to SILVA.
Comment 14: Figure 6. Please provide an isotype control for CD4+CD25+FoxP3+ and scatter plots for the rest of the antibody markers.
Response 14: These are now shown as Supplementary Figures 4-5.
Comment 15: Figure 8. The marker of obesity should be TC, TG, HDL, LDL. Rather than the indicators presented by the authors. Why did the author not detect TC, TG, HDL, LDL? It is suggested to supplement relevant data.
TC, TG, HDL, LDL are markers of cardiovascular health, insulin resistance and are good readouts for an obesity study. However they will all increase in response to a high fat diet which all animals in this study were exposed to. Our focus was on possible causes of the different fat accumulation phenotype in adipose tissue. Secondly the serum samples are more than 2 years old and may be unsuitable for analysis.
Comment 16: The format of the References should be checked (refer to Authors Guidelines).
All referances have been rechecked and formatted to be the same format.
Comment 1: Nutrients seem to not require Highlights. If necessary, please reduce the Highlights to 4-5 entries.
Response: Thank you for pointing this out. The highlight points have been merged into four key points.
Comment 2: Please do not divide the Abstract section into two paragraphs.
Response: The abstract has been modified to one paragraph.
Comment 3: Please provide the conditions for experimental animals, such as environmental temperature, humidity, light, etc.
Response: This information has been updated in the method section as “All mice were singly housed in controlled environmental conditions (22°C), 12-hour light-dark cycle in shoebox cages containing corncob bedding”
Comment 4: Why the author's 16S sequencing be divided into V2-4-8 and V3-6, 7-9 detection segments? There are repeated intervals in these paragraphs. How did the author merge the data?
Response: The 16S amplification kit of the manufacturers (Thermofisher Scientific) comes with two primers targeting of V2-4-8 and V3-6, 7-9 regions. The 16S amplification was done by PCR as per the manufacturer’s instructions to prepare the final library. The Ion Torrent platform generates single-end fastq files per sample.
Comment 5: Please convert all rpm to rcf or g.
Response 5: Done. Only one place had a centrifuge speed identified
Comment 6: Figure 1B and C. What does gm mean? Is it g per mouse? If so, simply write it as g.
Response 6: gm of Figure 1B and C is the abbreviation for gram. The information has been updated in the legends of Figure 1.
Comment 7: Why does the first paragraph of results need to be italicized? Does it have any special meaning?
Response 7: Thank you for pointing this out. The formatting issue has been corrected.
Comment 8: There are problems with gene analysis and data processing, which must be corrected. If calculated based on 2−ΔΔCT, the relative gene expression level of one of the groups in the results should be 1.
Response 8: The gene expression was recalculated, and Figure 9 was updated with new figures generated from the reanalysis of the gene expression by the 2−ΔΔCT method.
Comment 9: There should be spaces between units and numbers.
Response 9: All edits are done.
Comment 10: Please unify the “Fig.” or “Figure” in the manuscript.
Response 9: Done. All converted to Figure.
Comment 11: Please unify the form of P value, p, or P.
Response:11: Formatting issues have been addressed. All p are now P.
Comment 12: Please italicize the name of the genus level of the bacteria.
Response 12: Formatting issues have been addressed.
Comment 13: Figure 3F. The results of 16S sequencing can only be accurate to the genus level. How are species-level data derived?
Response 13: The sequencing data at the species level was identified from the MicroSEQ database v2013.1 a comprehensive resource for quality checked and aligned ribosomal RNA sequence data that is proprietary to Thermofisher Scientific for use after sequencing on the Ion Torrent platform. MicroSEQ identifies more species compared to SILVA.
Comment 14: Figure 6. Please provide an isotype control for CD4+CD25+FoxP3+ and scatter plots for the rest of the antibody markers.
Response 14: These are now shown as Supplementary Figures 4-5.
Comment 15: Figure 8. The marker of obesity should be TC, TG, HDL, LDL. Rather than the indicators presented by the authors. Why did the author not detect TC, TG, HDL, LDL? It is suggested to supplement relevant data.
TC, TG, HDL, LDL are markers of cardiovascular health, insulin resistance and are good readouts for an obesity study. However they will all increase in response to a high fat diet which all animals in this study were exposed to. Our focus was on possible causes of the different fat accumulation phenotype in adipose tissue. Secondly the serum samples are more than 2 years old and may be unsuitable for analysis.
Comment 16: The format of the References should be checked (refer to Authors Guidelines).
All referances have been rechecked and formatted to be the same format.
Comment 1: Nutrients seem to not require Highlights. If necessary, please reduce the Highlights to 4-5 entries.
Response: Thank you for pointing this out. The highlight points have been merged into four key points.
Comment 2: Please do not divide the Abstract section into two paragraphs.
Response: The abstract has been modified to one paragraph.
Comment 3: Please provide the conditions for experimental animals, such as environmental temperature, humidity, light, etc.
Response: This information has been updated in the method section as “All mice were singly housed in controlled environmental conditions (22°C), 12-hour light-dark cycle in shoebox cages containing corncob bedding”
Comment 4: Why the author's 16S sequencing be divided into V2-4-8 and V3-6, 7-9 detection segments? There are repeated intervals in these paragraphs. How did the author merge the data?
Response: The 16S amplification kit of the manufacturers (Thermofisher Scientific) comes with two primers targeting of V2-4-8 and V3-6, 7-9 regions. The 16S amplification was done by PCR as per the manufacturer’s instructions to prepare the final library. The Ion Torrent platform generates single-end fastq files per sample.
Comment 5: Please convert all rpm to rcf or g.
Response 5: Done. Only one place had a centrifuge speed identified
Comment 6: Figure 1B and C. What does gm mean? Is it g per mouse? If so, simply write it as g.
Response 6: gm of Figure 1B and C is the abbreviation for gram. The information has been updated in the legends of Figure 1.
Comment 7: Why does the first paragraph of results need to be italicized? Does it have any special meaning?
Response 7: Thank you for pointing this out. The formatting issue has been corrected.
Comment 8: There are problems with gene analysis and data processing, which must be corrected. If calculated based on 2−ΔΔCT, the relative gene expression level of one of the groups in the results should be 1.
Response 8: The gene expression was recalculated, and Figure 9 was updated with new figures generated from the reanalysis of the gene expression by the 2−ΔΔCT method.
Comment 9: There should be spaces between units and numbers.
Response 9: All edits are done.
Comment 10: Please unify the “Fig.” or “Figure” in the manuscript.
Response 9: Done. All converted to Figure.
Comment 11: Please unify the form of P value, p, or P.
Response:11: Formatting issues have been addressed. All p are now P.
Comment 12: Please italicize the name of the genus level of the bacteria.
Response 12: Formatting issues have been addressed.
Comment 13: Figure 3F. The results of 16S sequencing can only be accurate to the genus level. How are species-level data derived?
Response 13: The sequencing data at the species level was identified from the MicroSEQ database v2013.1 a comprehensive resource for quality checked and aligned ribosomal RNA sequence data that is proprietary to Thermofisher Scientific for use after sequencing on the Ion Torrent platform. MicroSEQ identifies more species compared to SILVA.
Comment 14: Figure 6. Please provide an isotype control for CD4+CD25+FoxP3+ and scatter plots for the rest of the antibody markers.
Response 14: These are now shown as Supplementary Figures 4-5.
Comment 15: Figure 8. The marker of obesity should be TC, TG, HDL, LDL. Rather than the indicators presented by the authors. Why did the author not detect TC, TG, HDL, LDL? It is suggested to supplement relevant data.
TC, TG, HDL, LDL are markers of cardiovascular health, insulin resistance and are good readouts for an obesity study. However they will all increase in response to a high fat diet which all animals in this study were exposed to. Our focus was on possible causes of the different fat accumulation phenotype in adipose tissue. Secondly the serum samples are more than 2 years old and may be unsuitable for analysis.
Comment 16: The format of the References should be checked (refer to Authors Guidelines).
All referances have been rechecked and formatted to be the same format.
Round 2
Reviewer 2 Report
Manuscript is improved now.
Need minor corrections
Reviewer 3 Report
The author answered all my questions well and I support publication of this manuscript in its present form.